# Innovative Approaches to Monitor Central Line Associated Bloodstream Infections (CLABSIs) Bundle Efficacy in Intensive Care Unit (ICU): Role of Device Standardized Infection Rate (dSIR) and Standardized Utilization Ratio (SUR)—An Italian Experience

**DOI:** 10.3390/jcm13020396

**Published:** 2024-01-11

**Authors:** Silvia Boni, Marina Sartini, Filippo Del Puente, Giulia Adriano, Elisabetta Blasi Vacca, Nicoletta Bobbio, Alessio Carbone, Marcello Feasi, Viviana Grasso, Marco Lattuada, Mauro Nelli, Martino Oliva, Andrea Parisini, Roberta Prinapori, Maria Carmela Santarsiero, Stefania Tigano, Maria Luisa Cristina, Emanuele Pontali

**Affiliations:** 1Department of Infectious Diseases, Galliera Hospital, 16128 Genoa, Italy; silvia.boni@galliera.it (S.B.); filippo.del.puente@galliera.it (F.D.P.); elisabetta.blasi@galliera.it (E.B.V.); nicoletta.bobbio@galliera.it (N.B.); marcello.feasi@galliera.it (M.F.); andrea.parisini@galliera.it (A.P.); roberta.prinapori@galliera.it (R.P.); stefania.tigano@galliera.it (S.T.); 2Operating Unit Hospital Hygiene, Galliera Hospital, 16128 Genoa, Italy; marina.sartini@galliera.it (M.S.); ale_carbo@live.it (A.C.); martinooliva@outlook.it (M.O.); maria.luisa.cristina@galliera.it (M.L.C.); 3Department of Health Sciences, University of Genoa, 16132 Genoa, Italy; 4Hospital Infection Control Committee, Galliera Hospital, 16128 Genoa, Italy; giulia.adriano@galliera.it (G.A.); maria.santarsiero@galliera.it (M.C.S.); 5Anaesthesia and Intensive Care Unit, E.O. Ospedali Galliera, 16128 Genoa, Italy; viviana.grasso@galliera.it (V.G.); marco.lattuada@galliera.it (M.L.); 6Medical Service Management, Galliera Hospital, 16128 Genoa, Italy; mauro.nelli@galliera.it

**Keywords:** device Standardized Infection Ratio (dSIR), central lines associated bloodstream infections (CLABSI), intensive care units, Standardized Utilization Rate (SUR)

## Abstract

In several settings, the COVID-19 pandemic determined a negative impact on the occurrence of healthcare-associated infection, particularly for on central lines associated bloodstream infections (CLABSI). In our setting, we observed a significant increase in CLABSI in our intensive care unit (ICU) during 2020 and 2021 vs. 2018 to 2019. A refresher training activity on central venous catheter (CVC) management bundles was carried out in September–October 2021 for the ICU health staff. We assessed the impact of bundle implementation by means of standardized indicators, such as the Device Utilization Ratio (DUR), in this case, the Central Line Utilization Ratio, the Standardized Utilization Ratio (SUR), and the device Standardized Infection Ratio (dSIR). Standardized ratios for device use and infection ratio were computed using data from 2018 and 2019 as expectation data. After bundle implementation, we observed a significant reduction of dSIR (*p* < 0.001), which dropped from 3.23 and 2.99 in the 2020–2021 biennium to 1.11 in 2022 (CLABSI in the first quarter only); no more CLABSI were observed afterwards. Standardized ratios proved helpful in identify increasing trends of CLABSI in the ICU and monitoring the impact of a simple effective tool, i.e., training on and implementation of a bundle for CVC management.

## 1. Introduction

Healthcare-associated infections (HAIs) pose a significant challenge in modern healthcare, contributing to increased morbidity, mortality, and healthcare costs [1,2]. These infections have a profound impact on intensive care units (ICUs), where patients are often critically ill and more susceptible to infections [3].

Antimicrobial stewardship programs and surveillance activities have emerged as key strategies in managing HAIs and improving patients’ outcomes by optimizing antimicrobial use, reducing microbial resistance, and decreasing the spread of infections caused by multidrug-resistant organisms (MDROs) [4]. Previous research has demonstrated both positive and neutral impacts of antimicrobial stewardship on HAIs incidence and prevalence [5].

Surveillance programs for HAIs are an essential component in monitoring HAIs incidence. Furthermore, by promptly identifying the extent and characteristics of an outbreak, such programs may reduce the subsequent incidence of HAIs [6]. The impact of outbreaks and ongoing diffusion of HAIs could be efficiently contrasted by implementing specific bundles (i.e., structured evidence-based procedures typically comprising three to five components that, when executed collectively and consistently, have been demonstrated to enhance patient outcomes) [7]. 

Bundles have proven to be effective in clinical practice for the purpose of preventing central lines associated bloodstream infections (CLABSIs). In 2006, Provonost et al. published an evidence-based intervention that resulted in a large and sustained reduction in CLABSIs in ICU [8]. During recent years, several authors from different countries reported similar results, thus confirming the role of such bundles [7,9,10,11]. However, adherence to bundles in antimicrobial stewardship is challenging to measure [12,13]. Despite these difficulties, several studies have shown adherence to these bundles could significantly improve patient outcomes [7,9,11,14,15,16].

In several settings, the COVID-19 pandemic had a negative impact on the adherence of healthcare staff to infection control measures and on circulation of MDROs in the same settings. This is evident from several reports from a large number of countries [17,18,19,20,21,22,23,24]. As known, the strategies to prevent CLABSI are linked to correct insertion and maintenance of the device [25]. Actually, as previously mentioned, several infection control strategies were (partially or largely) neglected during the pandemic, in view of the high number of patients admitted in serious conditions, within a short time frame and during phases of experienced staff shortage and increased turnover of both patients and staff [24]. It should also be pointed out Italy was, chronologically, the second nation to undergo the early impact of the COVID-19 pandemic. During the early stages of the pandemic, the effectiveness of treatments lacked high-level scientific evidence, resulting in frequent negative outcomes for many patients [26]. In this critical situation, the adequate management techniques for catheters, such as the maintenance of line dressing integrity and the hub scrub with chlorhexidine, received less attention due to the increased workload associated with critically ill COVID-19 patients.

In our setting, we observed an apparent increase in the incidence of CLABSI in ICU where a large proportion of patients were affected by COVID-19, thus it was decided to study the phenomenon and make specific interventions targeting the correct implementation of bundles for central vein catheters management.

Thus, the objective of our study is to assess, through the Standardized Infection Rate (SIR) and Standardized Utilization Ratio (SUR), the effectiveness of the bundles in decreasing the escalated occurrence of CLABSI events during the COVID-19 pandemic period in a single Italian ICU [27].

## 2. Results

We observed a total of 1679 admissions to ICU over a 5-year period (from 1 January 2018 to 31 December 2022). The number of admissions per year was not stable as it declined from the first, pre-COVID-19, period (2018–2019) to the second, pandemic, period (2020–2021). This reduction was the result of a significant decrease in surgical procedures and the necessity to admit patients affected by severe COVID-19 pneumonia to the ICU. The overall duration of patients’ stays in the ICU increased from the pre-COVID-19 to pandemic period (Table 1). Actually, the different type of patients admitted to the ICU determined an increase in the average length of stay of patients from the first (6.09 and 6.03 days) to the second period (7.84 and 8.45 days).

Moreover, following the onset of the COVID-19 pandemic, we observed an increase in the use of central lines (Table 1). Although incidence of SARS-CoV-2 infections significantly decreased in 2022, the use of central lines remained high, while the number of patients treated in 2022, as well as total stay of patients in ICU, were similar to the 2020–2021 biennium.

CLABSI incidence increased from the first period to the second one; this was followed by a drastic reduction in 2022 (63% less than 2021) (Figure 1). CLABSI incidence significantly increased from the pre-pandemic years 2018–2019 to the intra-pandemic years 2020–2021 (*p* < 0.01 chi-square test).

The average time between central venous catheter (CVC) insertion and the onset of CLABSI was 12 days. This timing did not change over the study period, with the exception of two cases of CLABSI, which occurred shortly after CVC implantation.

In light of the persistence of a relatively elevated incidence of CLABSIs during 2022, a quarter-by-quarter assessment was carried out starting from the first quarter of 2020. The July–September 2021 quarter was not included because all training activities were carried out during it, and there could have been a bias in performance. This analysis revealed this persistently increased incidence was due to several events occurring in the first quarter of 2022. Subsequently, the incidence reached zero and remained stable during the following three quarters (Figure 2).

Regarding CVC utilization, a variable rate of intravascular device utilization was observed. When compared to the pre-COVID period (2018–2019), DUR was lower, and remained stable over the pandemic years. This could lead to the assumption that the device use was lower during the pandemic. This reduction was especially pronounced in the second quarter of 2021, reaching 0.526 (Figure 3).

However, when adjusted for the aforementioned factors, and therefore calculating the correspondent standardized value (SUR), we observed the use of CVC remained stable during the pandemic period when compared to the pre-COVID-19 period, with values just slightly different from 1.0. Following the end of the pandemic period, during 2022, and despite the reduction of patients hospitalized in the ICU, the SUR figures rose to 1.47, demonstrating a clear increase in CVC utilization when adjusted for the correct variables (Figure 4). 

After acquiring the standardized measurement of CVC utilization, we undertook the computation of the trend in the occurrence of CLABSI. This analysis was adjusted for the utilization of the device, thereby enabling us to derive the dSIR value. The dSIR values, not surprisingly, exceeded the projected value, reaching 3.23 in 2020 and 2.99 in 2021. Nevertheless, the decline detected through incidence analysis in 2022 was inadequate to attain a value below 1.0. Even though it stood at 1.11, marginally surpassing the anticipated pre-pandemic values. Nevertheless, the difference between the pandemic period and 2022 reached a statistically significant variation (*p* < 0.001). (Figure 5).

The population SIR (pSIR) was subsequently calculated using the dSIR to assess the CLABSI trend through a population analysis. Even in this particular case, the pSIR values (which were determined based on both the dSIR and SUR) were higher throughout the duration of the pandemic (3.42 in 2020, 2.77 in 2021), with a decrease observed in 2022, albeit to a lesser extent than that calculated using the dSIR (1.64) (Figure 6). This may be due to the nature of our intervention, as we designed it to ameliorate the management of the CVC, not to reduce its use.

### Intervention

We directly observed device insertions twice with 100% adherence to the insert device bundle. We decided to observe a limited number of insertions because our retrospective evaluation of CLABSI identified a really low number of early ones. Differently, since the median time to CLABSI onset was of 12 days, we assumed the major difficulties were in CVC management rather than insertion. Thus, we observed more frequently CVC management. Observations of CVC management occurred six times, with registration of observation or lack of observation of the different bundle items. In particular, the items that were most adhered to were “the intact dressing and replacing every 7 days” (100%), while those that were least adhered to were “hub scrub with 2% chlorhexidine in 70% isopropyl alcohol” (33.33%). The observations evaluated the adherence to each single bundle point; 1 point was given for each bundle that was respected, 0 point if the bundle was not respected. The overall bundle adherence rate during observation visits was 78.12%.

## 3. Discussion

COVID-19 has caused an increase in the complexity of clinical management of patients in ICU, thus determining both an increase in resources used and in patient stay [28,29,30]. The surveillance of the incidence of CLABSI revealed a negative impact of SARS-CoV-2 pandemic also in our setting, resulting in a statistically significant increase in CLABSI incidence in our ICU.

A lot of studies identified several potential factors that could have affected the observed increased circulation of MDROs and increased incidence of CLABSI and other infections in the ICU during COVID-19 pandemic. Some factors, although not all unanimously confirmed, may have also contributed to the observed increase in patient complexity and length of stay in ICU. They include, for example, overcrowding of intensive care units, prolonged and repeated patient pronation, improper/inadequate use of protective devices, alteration in infection control procedures, treatment of immunomodulating agents (e.g., tocilizumab, sarilumab, anakinra, etc), longer ICU stays, recruitment of untrained personnel, and even in the incidence of infections caused by MDROs [24,27,31,32,33,34,35,36,37,38,39,40]. In particular, the activity of proning patients to improve their respiratory function by improving lung recruitment and better lung ventilation/perfusion matching has proven to be an important determinant, significantly associated with increased colonization by Carbapenem-Resistant Enterobacteriaceae (CRE), device loss or traction, and bacteremia [27,31,35,41,42,43,44]. The increased rate of intravascular device use reported with the pandemic onset could lead to the conclusion that the increased incidence of CLABSI may be due not only to the increased complexity of patients admitted to the ICU during the COVID-19 pandemic, but also to the lack of adherence to infection control measures. Actually, the observed overcrowding in the ICU, together with trained staff shortages, may have caused a reduction in the frequency of contacts with patients and in CVC maintenance (e.g., chlorhexidine bathing, scrubbing the hub, site examinations) as well as disruptions in processes of care (e.g., risking disrupting catheter dressings when placing patients in a prone position), thus contributing to an increase in CLABSI incidence in this setting [27,34,37,45]. Another significant aspect was the changed feature of patients hospitalized in the ICU. During the COVID-19 pandemic, at least initially, admission policies in ICUs changed due to limitations in major surgery activities, thus privileging COVID-19 patients with consequent limited access to less severe cases. Thus, the increase in CLABSI rates could be due to the decrease in the denominator, which was primarily composed of patients with lower CLABSI risk [34,37].

In our experience, amidst the COVID-19 pandemic, we came across multiple potentially significant events in the ICU. In particular, we observed a decline in the number of hospitalized patients, an increase in the average duration of patients’ stays, and a surge in the utilization of CVCs. In our experience, average patients’ ICU stays exhibited a decline in 2022, following the reduction of COVID-19 pneumonia cases, while CVC use remained elevated. Concerning CLABSI incidence, we observed a significant rise in incidence during the pandemic period. This situation, in September 2021, led to the training interventions designed to improve the CVC management. Subsequently, a ‘refreshed’ implementation of CVC bundles was carried out. Following the intervention, the occurrence of CLABSIs remained stable for two quarters, and then receded after the second quarter of 2022.

To standardize the data and mitigate confounding biases associated with different patient types and therapeutic strategies related to the insertion of the CVC across distinct time periods (pandemic and post-pandemic), we have conducted an analysis utilizing unstandardized (DUR) and standardized ratios (SUR, dSIR, and pSIR). The DUR, which represented the device utilization ratio, revealed a consistent or potentially decreased utilization of CVC in comparison to the reference period of 2018–2019 (serving as a benchmark for our prediction).

The SUR, nonetheless, ascertained, upon standardization, the data revealed considerable stability in CVC use, for which we observed an increase in 2022 as a result of the deflating of the pandemic. The dSIR and the pSIR identified a noteworthy decrease in the occurrence of CLABSI, even after standardization for the device’s utilization and the category of hospitalized patient, thereby demonstrating CLABSI reduction was not associated with different categories of patients or with reduced device use.

We can conclude our study showed the potential for rapid favourable outcomes associated with the utilization of bundles in ICU that target CVC management. Our study confirms existing data in scientific literature concerning bundles’ efficacy in reducing CLABSI incidence [16,46,47]. Nonetheless, our study has additionally provided valuable insights derived from standardized data, including the incidence of CLABSI, the components encompassed within the bundle, and the level of compliance exhibited by healthcare personnel towards adhering to these components. Our application of standardized ratios for the analysis of incidence has additionally facilitated the alleviation of selection bias effects, allowing for a more precise analysis compared to a relatively raw incidence analysis.

### 3.1. Limitations

The main limitations of this study is that the investigation was carried out in a single ICU, only. Other limitations include the observational and retrospective nature of the study, including data collection and evaluation of the impact of the training activity. Such limitations may have had a limited impact since the main epidemiological findings were coherent with current literature. In addition, the impact of training activity and of the subsequent implementation of CVC bundles is in accordance with what is reported in the literature in the pre-COVID-19 era.

### 3.2. Future Implications

The exercise of collecting local epidemiological data is relatively easy, but the information from raw data can be misleading. The use of standardized ratios helped us to properly compare data from quarter to quarter, from year to year, independently from the changing epidemiological landscape and patient types in the ICU. The use of standardized ratios may help in comparing epidemiological findings with other regional, national, or international data/studies. Their diffusion and their sharing in scientific publications could be really helpful in comparing our findings with data from other settings.

Other messages for the future are to keep looking at the basics of infection control and to remember that key simple interventions such as training, bundles implementations, and monitoring can be extremely cost-effective.

## 4. Materials and Methods

The study is a retrospective evaluation of CLABSI incidence in a northern Italian nationally renowned and highly specialized hospital organized by treatment intensity. The aforementioned hospital consists of three buildings and accommodates 458 beds, primarily in 3- and 4-bed rooms, with over 15,000 routine admissions annually, along with more than 8600 medical procedures conducted in outpatient and day surgery settings. The ICU within the hospital is designed as an open space, incorporating two isolation rooms, collectively offering 8 beds. Furthermore, the ICU was equipped with the capability to expand its capacity to accommodate up to 12 beds, as it frequently happened during COVID-19 epidemic peaks.

### 4.1. Population

We included all hospitalized patients in the ICU who had an episode of CLABSI from 1 January 2018 to 31 December 2022 (5 years of observation).

The CLABSI events were identified according to CDC definition, and the data were collected, including device, patients’ days and microorganisms’ infection-related data [48]. Only CLABSI attributable to the ICU, in accordance with CDC criteria, were analyzed.

We calculated the incidence of CLABSI (CLABSI number/line days) by dividing it into both quarters and years. Analyses were performed for both cases. To give a better understanding of the intervention’s efficacy, the Device Utilization Ratio (DUR), in this case, the Central Line Utilization Ratio (DUR), the Standardized Utilization Ratio (SUR), and the device Standardized Infection Ratio (dSIR) were used as tools for analyzing trends of HAIs [49,50]. These metrics provide valuable insights into the incidence and prevalence of HAIs, aiding in the evaluation and improvement of infection control measures. 

As the DUR defines but does not standardize the degree of device utilization, SUR is theoretically more informative, as it standardizes the measure obtained with DUR by adjusting it for various facility and/or location-level factors that contribute to device use [50]. On the other hand, dSIR is a standardized measure used to track HAIs at a national, state, or local level over time and can be used to measure progress from a single point in time [49]. Values obtained are normally compared with benchmarks (in our case, are based on the expectations from the previous data of the ICU during the 2018–2019 biennium). A result greater than 1.0 indicates the events were superior to what is predicted; conversely, a value inferior to 1.0 indicates events were fewer than predicted.

A key component of device risk reduction is decreasing exposure to the device, either by preventing its insertion or reducing its duration of use. By examining both dSIR (reflecting rate of infection and device use) and SUR (reflecting device use), the impact of interventions can be measured. It should be noted changes in dSIR may also occur if the interventions result in a significant difference in catheter use and frequently underestimate improvement in infection rates, mostly because they fail to account for reduced device utilization associated with infection prevention intervention. To overcome this problem, Fakih MG et al. in their 2019 paper proposed the use of the population SIR (pSIR), which combined the device SIR (dSIR) and the standardized device use ratio (SUR) [51]. Thus, the pSIR refers to the entire population, adjusted for expected device use. The value of the pSIR is calculated as dSIR*SUR.

### 4.2. Description of Training on CVC Bundles

All ICU staff were trained with a specific course on HAIs, including formal lectures on principles of HAIs and specific bundles, and hands-on practice on a training dummy. 

Specific training activities targeting CVCs in our program were based on the introduction and application in clinical practice of bundles regarding actions ranging from the insertion to the management of CVCs. A newly developed manual for bundles at our institution (Available at: https://www.galliera.it/20/58/strutture-e-servizi-in-staff-alla-direzione-sanitaria/858/io-e-manuali-cio/manuale-operativo-mo/bundle-manuale-per-la-prevenzione-delle-ica/view, (accessed on 19 December 2023)) was used for such training activities. The key issues included in the manual and discussed during the training were:device insertion,guided ultrasound procedure,surgical hand washing aseptic technique,skin antisepsis with 2% chlorhexidine in 70% isopropyl alcohol,use of sutureless fixation device,management,hand washing with alcohol solution before and after using the catheter,hub scrub with chlorhexidine,keep the dressing intact and replace it every 7 days,remove as soon as possible.

In addition, washable keepsake posters based on the new bundles’ manual were developed and affixed throughout the hospital, including the ICU. 

Based on CLABSI surveillance data from 2020 and first quarters of 2021, improvement actions were planned and implemented from September 2021 to October 2021. The training activities involved 44 nurses over one month. 

Ten safety walks were carried out in the ICU to promote culture safety and raise awareness among operators regarding the rise in CLABSI incidence.

Starting from January 2022, six direct observations were carried out to evaluate bundles’ adherence. 

### 4.3. Statistical Analysis

All patient characteristics are presented as mean with standard deviation, median, and range for continuous variables, and expressed as absolute values along with percentages for categorical variables. The Chi-squared test was used to assess independence between variables. A *p*-value of less than 0.05 was considered statistically significant. All statistical analyses were performed using Stata/SE 18 software (StataCorp LP, College Station, TX, USA).

## 5. Conclusions

When faced with the COVID-19 pandemic and the increase in difficulties associated with patient management, coupled with the subsequent surge in CLABSI incidence in the ICU, conducting an investigation based on standardized information and the implementation of bundled interventions held the capacity to detect a problem and elicit a relatively swift response aimed at curtailing the incidence of CLABSI.

Most of our findings were in line with the international literature. However, the key aspect of our study was that we always needed to go beyond our, even scientifically sound, observations. We need to find solutions to the detected problems, and we proved that sometimes relatively easy solutions existed to overcome worrisome situations. In addition, we proved, by means of standardized indicators, the effectiveness of the introduced intervention. Investing in people (training on bundles) showed to be extremely effective in reducing life-threatening infections (CLABSIs in ICU). 

## Figures and Tables

**Figure 1 jcm-13-00396-f001:**
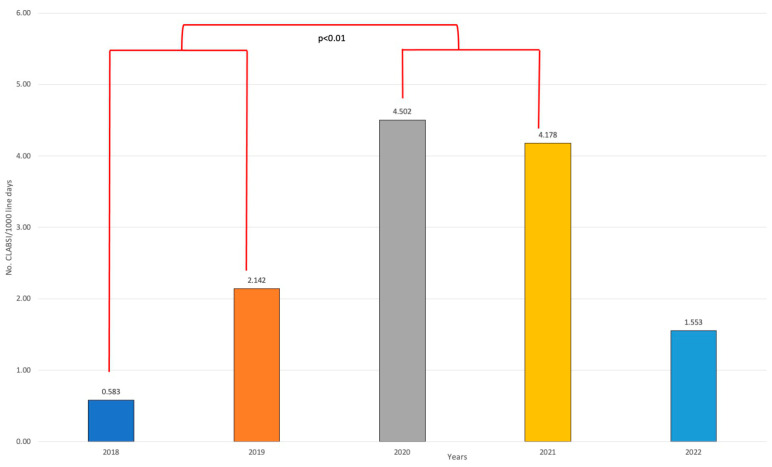
Central lines associated bloodstream infections (CLABSI) incidence per year during study period.

**Figure 2 jcm-13-00396-f002:**
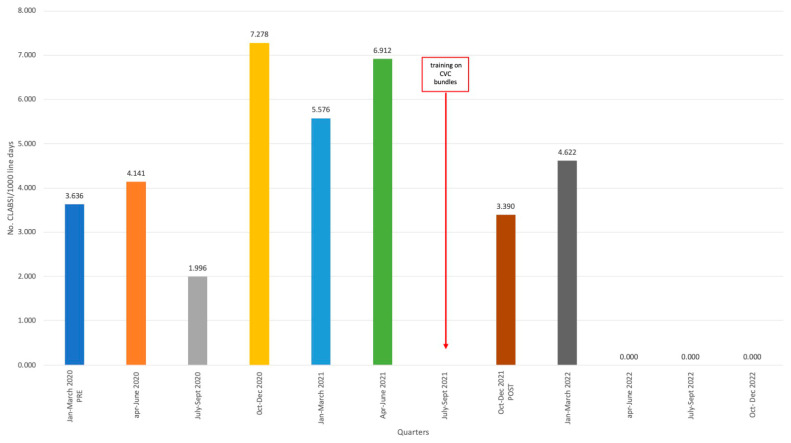
Incidence of Central lines associated bloodstream infections (CLABSI) broken down by quarters from 2020 to 2022.

**Figure 3 jcm-13-00396-f003:**
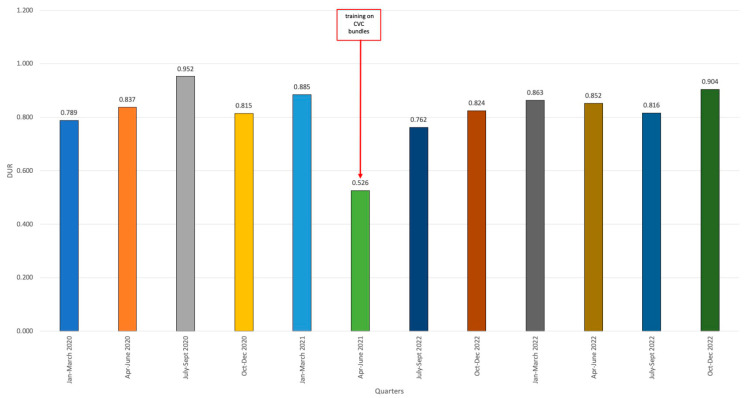
Device Utilization Ratio (DUR) values by quarters between the last quarter of 2020 and the last quarter of 2022.

**Figure 4 jcm-13-00396-f004:**
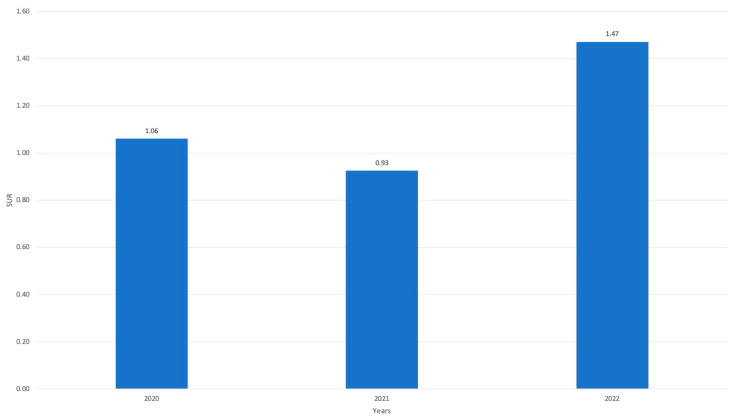
Standardized Utilization Ratio (SUR) values during 2020, 2021, and 2022 in comparison with 2018 and 2019.

**Figure 5 jcm-13-00396-f005:**
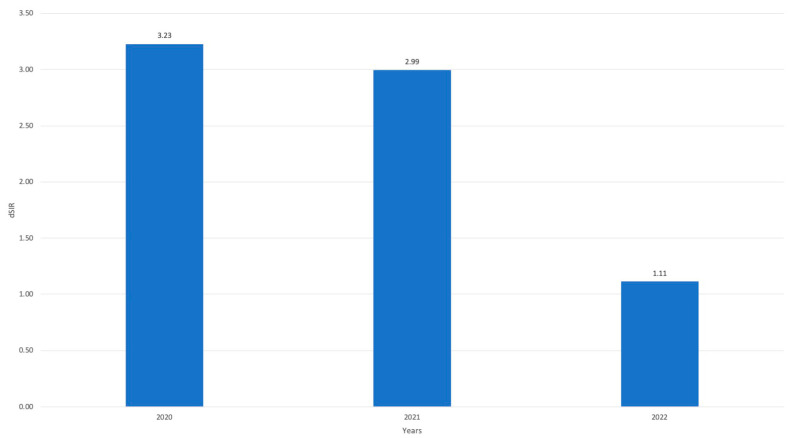
Standardized Infection Rate (SIR) adjusted for the utilization of CVC (dSIR) calculated over the three-year period 2020–2022, based on the expected figures estimated from 2018–2019.

**Figure 6 jcm-13-00396-f006:**
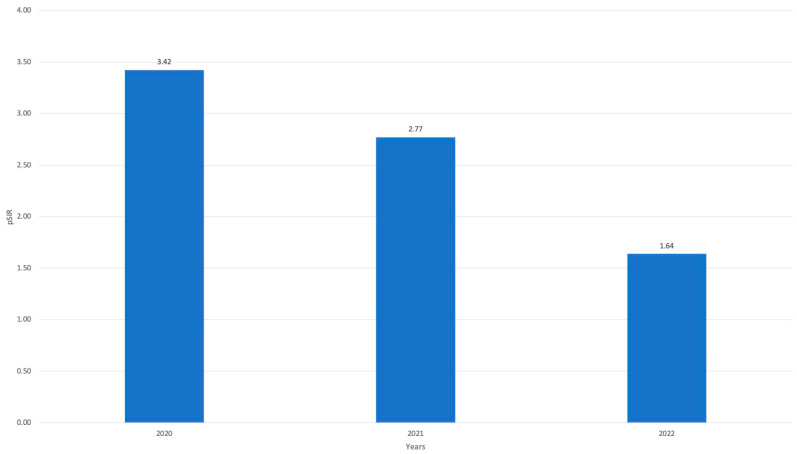
Population SIR (pSIR) by year from 2020 to 2022.

**Table 1 jcm-13-00396-t001:** Number of patients, total intensive care unit (ICU) days, and total central line days by year of study surveillance.

Year	Number of Patients	Patients Stay in ICU (Total Days)	Central Line Days
2018	375	2284	1716
2019	371	2238	1867
2020	337	2643	2221
2021	309	2613	1915
2022	287	2252	1932

## Data Availability

The data presented in this study are available on request from the corresponding author. The data are not publicly available due to internal regulations.

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
