# Peer review of "Innovative Approaches to Monitor Central Line Associated Bloodstream Infections (CLABSIs) Bundle Efficacy in Intensive Care Unit (ICU): Role of Device Standardized Infection Rate (dSIR) and Standardized Utilization Ratio (SUR)—An Italian Experience"

_jcm, 2024, doi:10.3390/jcm13020396_

Round 1
Reviewer 1 Report
Comments and Suggestions for Authors
The authors conducted an interesting evaluation of the role of dSIR and SUR to monitor CLABSIs bundle efficacy in ICU. Thank you for your submission.
In general, the manuscript has been well-written and -conducted. I have just a few minor recommendations.
- Title: This would be important to make it clear that the study has been conducted in Italy, as those results are all from an Italian reality and panorama.
- Abstract: The authors should include the statistical approach used in the study. Also, I suggest that the authors include where their data were obtained from. Please include it in this sentence: “Our objective was to assess the effectiveness on central lines associated bloodstream infections (CLABSI) incidence of the implementation of prevention central venous catheter (CVC) management bundle based on the Standardised Infection Rate (SIR) and Standardised Utilisation Rate (SUR) indicators in an intensive care unit (ICU)”
- Introduction: This section is very well-constructed, using up-to-date references and introducing the problem adequately.
“Unfortunately, the COVID-19 pandemic had a negative impact”. Unfortunately is too personal. Please start the sentence from “The COVID-19…”.
Also, similarly to the abstract, please include that this study was conducted using data specifically from Italy in this phrase “The objective of our study is to assess, through the Standardized Infection Rate (SIR) and Standardized Utilization Ratio (SUR), the effectiveness of the bundles in decreasing the escalated occurrence of CLABSI events during the COVID-19 pandemic period in a single ICU [17].”
- Figures, table, and captions: The quality of the figures is not satisfactory. They need to be improved in terms of resolution. Also, the axes should be carefully and correctly named according to their meaning. In addition, no caption has been noted for any figure or table; the authors should include it by stating important information such as the statistical method used, acronyms, etc. Regarding the table, the authors should be careful with its structure and revise it accordingly; for instance: this should not be all closed. Please revise these aspects.
- Conclusion: Please revise it and draw a conclusion linked to the study objectives and results found.
Author Response
The authors conducted an interesting evaluation of the role of dSIR and SUR to monitor CLABSIs bundle efficacy in ICU. Thank you for your submission.
In general, the manuscript has been well-written and -conducted.
Thanks for your positive feedback
I have just a few minor recommendations.
- Title: This would be important to make it clear that the study has been conducted in Italy, as those results are all from an Italian reality and panorama.
Title has been modified to accommodate this suggestion
- Abstract: The authors should include the statistical approach used in the study. Also, I suggest that the authors include where their data were obtained from. Please include it in this sentence: “Our objective was to assess the effectiveness on central lines associated bloodstream infections (CLABSI) incidence of the implementation of prevention central venous catheter (CVC) management bundle based on the Standardised Infection Rate (SIR) and Standardised Utilisation Rate (SUR) indicators in an intensive care unit (ICU)”
Abstract was revised
- Introduction: This section is very well-constructed, using up-to-date references and introducing the problem adequately.
“Unfortunately, the COVID-19 pandemic had a negative impact”. Unfortunately is too personal. Please start the sentence from “The COVID-19…”.
Modified as recommended
Also, similarly to the abstract, please include that this study was conducted using data specifically from Italy in this phrase “The objective of our study is to assess, through the Standardized Infection Rate (SIR) and Standardized Utilization Ratio (SUR), the effectiveness of the bundles in decreasing the escalated occurrence of CLABSI events during the COVID-19 pandemic period in a single ICU [17].”
Modified as recommended
- Figures, table, and captions: The quality of the figures is not satisfactory. They need to be improved in terms of resolution.
Included figures have the resolution requested by the Editorial rules
Also, the axes should be carefully and correctly named according to their meaning. In addition, no caption has been noted for any figure or table; the authors should include it by stating important information such as the statistical method used, acronyms, etc.
Figures were carefully revised. Axes with missing names received a name.
Regarding the table, the authors should be careful with its structure and revise it accordingly; for instance: this should not be all closed. Please revise these aspects.
Table 1 has been carefully revised and modified according to suggestion
- Conclusion: Please revise it and draw a conclusion linked to the study objectives and results found.
Conclusions were revised
Reviewer 2 Report
Comments and Suggestions for Authors
The results are presented in a clear manner, but the abstract lacks specific numerical values or percentages regarding the reduction in standardized infection ratios. Including these details would provide a more concrete understanding of the observed changes.
The rationale for using bundles is well explained, with a specific reference to their proven effectiveness in preventing central lines associated bloodstream infections (CLABSIs). The citation of Provonost et al.'s work in 2006 adds historical context but could benefit from more recent evidence.
The transition from the background information to the study's objective is smooth. However, the introduction could benefit from a concluding sentence that bridges the gap between the background and the specific focus on CLABSI during the COVID19 pandemic.
Definition of Standardized Ratios: The clear definition of DUR, SUR, dSIR, and pSIR enhances the understanding of how these ratios contribute to the evaluation of infection control measures. Statistical Analysis: The statistical analysis plan is appropriately outlined, with clear information about how patient characteristics will be presented and the statistical tests to be employed. The use of Stata/SE 18 software is a standard choice for statistical analysis.
Limitations and Transparency-The section on limitations is missing, and it's important to acknowledge potential biases or constraints related to the retrospective and observational nature of the study. A brief discussion of limitations would contribute to the transparency of the research. Future Implications: Consider incorporating a brief mention of potential implications for future research or healthcare practices. This could provide a forward-looking perspective and highlight areas for further investigation or improvement.
Author Response
The results are presented in a clear manner, but the abstract lacks specific numerical values or percentages regarding the reduction in standardized infection ratios. Including these details would provide a more concrete understanding of the observed changes.
Abstract was revised
The rationale for using bundles is well explained, with a specific reference to their proven effectiveness in preventing central lines associated bloodstream infections (CLABSIs). The citation of Provonost et al.'s work in 2006 adds historical context but could benefit from more recent evidence.
New (recent) citations from 2016, 2020 and 2021 have been added
The transition from the background information to the study's objective is smooth. However, the introduction could benefit from a concluding sentence that bridges the gap between the background and the specific focus on CLABSI during the COVID19 pandemic.
The recommended sentence has been added
Definition of Standardized Ratios: The clear definition of DUR, SUR, dSIR, and pSIR enhances the understanding of how these ratios contribute to the evaluation of infection control measures.
Thanks for your positive feedback
Statistical Analysis: The statistical analysis plan is appropriately outlined, with clear information about how patient characteristics will be presented and the statistical tests to be employed. The use of Stata/SE 18 software is a standard choice for statistical analysis.
Thanks for your positive feedback
Limitations and Transparency-The section on limitations is missing, and it's important to acknowledge potential biases or constraints related to the retrospective and observational nature of the study. A brief discussion of limitations would contribute to the transparency of the research.
Limitation section was added
Future Implications: Consider incorporating a brief mention of potential implications for future research or healthcare practices. This could provide a forward-looking perspective and highlight areas for further investigation or improvement.
A specific section was introduced at the end of discussion.